# Fine-Resolution Population Mapping from International Space Station Nighttime Photography and Multisource Social Sensing Data Based on Similarity Matching

**Luyao Wang [1,2]**, **Hong Fan [1,2,\*]** and **Yankun Wang [3,4,5]**

[1]  State Key Lab for Information Engineering in Surveying, Mapping and Remote Sensing, Wuhan University, 129 Luoyu Road, Wuhan 430079, China
[2]  Collaborative Innovation Center of Geospatial Technology, Wuhan University, Wuhan 430079, China
[3]  Research Institute for Smart Cities, School of Architecture and Urban Planning, Shenzhen University, Shenzhen 518061, China
[4]  Shenzhen Key Laboratory of Spatial Information Smart Sensing and Services, Shenzhen 518061, China
[5]  Guangdong Key Laboratory of Urban Informatics, Shenzhen 518061, China
\*  Correspondence: hfan3@whu.edu.cn; Tel.: +027-6877-8475

**Abstract:** Previous studies have attempted to disaggregate census data into fine resolution with multisource remote sensing data considering the importance of fine-resolution population distribution in urban planning, environmental protection, resource allocation, and social economy. However, the lack of direct human activity information invariably restricts the accuracy of population mapping and reduces the credibility of the mapping process even when external facility distribution information is adopted. To address these problems, the present study proposed a novel population mapping method by combining International Space Station (ISS) photography nighttime light data, point of interest (POI) data, and location-based social media data. A similarity matching model, consisting of semantic and distance matching models, was established to integrate POI and social media data. Effective information was extracted from the integrated data through principal component analysis and then used along with road density information to train the random forest (RF) model. A comparison with WordPop data proved that our method can generate fine-resolution population distribution with higher accuracy ($R^2 = 0.91$) than those of previous studies ($R^2 = 0.55$). To illustrate the advantages of our method, we highlighted the limitations of previous methods that ignore social media data in handling residential regions with similar light intensity. We also discussed the performance of our method in adopting social media data, considering their characteristics, with different volumes and acquisition times. Results showed that social media data acquired between 19:00 and 8:00 with a volume of approximately 300,000 will help our method realize high accuracy with low computation burden. This study showed the great potential of combining social sensing data for disaggregating fine-resolution population.

**Keywords:** population mapping; ISS photography; point of interest data; location-based social media data; semantic matching; distance matching

## 1. Introduction

The rapid world population growth has brought a wide range of environment issues, such as carbon emission, waste treatment, resource shortage, and land destruction [1,2]. Accurate and high-resolution population distribution information is crucial for resource allocation, land use policy, environment protection, disaster prevention, and transportation planning [3–5].

Considering the importance of population information, most countries have formulated termly census surveys and published the data through the Census Bureau [6,7]. Census data have been widely adopted in the study of social science because of their high reliability and availability [8–10]. However, census survey is often conducted every 10 years or more due to its time-consuming and labor-intensive process. Moreover, the data provided by the census survey are coarse, mainly in the province, city, or town level, which limits their use in accurate studies [11]. To meet the requirements of mesoscale and microscale studies, researchers have explored approaches in disaggregating census population data into fine-scale grids by using new data sources. Remote sensing data, given their high resolution and short acquisition cycle, have been widely used in population mapping studies [12–14]. Two categories of remote sensing data are mainly used in previous studies: Daytime land observation and nighttime light (NTL) data.

Researchers believe that directly gaining the population distribution information is difficult, considering the high mobility of the population [15]. However, the population can be indirectly estimated on the basis of the distribution of houses along with the average resident population per house, which is often provided by the local government. Hence, daytime land observation data are used to provide building distribution data [16–18]. The building level information and vacancy rate information are also considered in subsequent studies to improve the estimation accuracy [19]. However, the population distribution has great spatial heterogeneity, especially in some developing countries, indicating that the population size may be small in cities with large residential areas; this phenomenon is common in Chinese western cities. By contrast, some cities are particularly crowded, such as Hong Kong and Beijing, where tens of millions of people live in limited housing resources [20,21]. Therefore, the method can only be used in underdeveloped regions without census data or other external data sources, but it cannot be adopted when a highly accurate population distribution information is required.

The rapid development of nighttime light sensors provides new insight to the problem. NTL data, which records the light intensity produced by anthropogenic activities, are proven effective in numerous urban-related and environmental issues, such as revealing urban development level, distinguishing urban expansion direction, and identifying urban hotspots [22–24]. Their superiorities in describing global scale urban development at pixel level and low acquisition cost make them suitable for population mapping. Through the previous attempts in estimating population density and population distribution using NTL data, high relevance was found between nighttime light and city-level population, which makes NTL data a reliable proxy for population mapping in province-level and city-level [25,26]. The Defense Meteorological Satellite Program/Operational Linescan System (DMSP/OLS) with long period (1992–2013) is the most widely used NTL data source and has been adopted by numerous researchers in population mapping [14,27,28]. However, the coarse resolution (30 arc seconds, at nadir) and serious blooming effect (digital number: 0–63) restrict the analysis accuracy [29]. To overcome the limitations of DMSP, another NTL sensor, namely, Suomi National Polar-orbiting Partnership (NPP) satellite with the Visible Infrared Imaging Radiometer Suite (VIIRS), was launched by the National Oceanic and Atmospheric Administration/National Geophysical Data Center in 2011 [30]. NPP/VIIRS, the new-generation NTL data, has higher spatial and radiometric resolution (15 arc seconds, 0.5 km by 0.5 km) and more radiometric detection range than DMSP/OLS [31,32]. Thus, the NPP/VIIRS data have been successfully used in various aspects, such as social economics, urban expansion, energy consumption, marine pollution, and atmospheric pollutant dispersion [31,33–36]. Although NPP/VIIRS data have been proven to be considerably superior to DMSP/OLS data in most fields, few yearly NTL products (seven years, from 2012 to present), even less than a census cycle (often 10 years in most countries), limit their use in population mapping or estimation. Therefore, no reliable census data are available for analysis using NPP/VIIRS data. The coarse DMSP/OLS data are still used in population mapping analysis by numerous studies [13,37]. In addition, NTL emissions depend on affluence and economic structure. NTL emissions produced by commercial advertising, construction site, and security lighting cannot directly reflect population distribution in small scale, due to the lack

of direct human activity information. Thus, combining external data sources in the analysis is essential if further accurate results are required. Current studies focusing on population mapping with DMSP data can be distinguished into three categories in accordance with the auxiliary data sources used: The first category is without extra data source. In these studies, only DMSP/OLS and census data are used [13,14,37]. The main goals are to model the relationship between NTL and census data through simple linear regression, least squares linear (OLS) regression, and geographically weighted regression (GWR), considering the spatial heterogeneity in county or town level, and to conduct the trained model in grid level for disaggregating census data. The second category is with daytime remote sensing data [38–41]. Realizing that the population mapping accuracy is unsatisfied, researchers attempt to use daytime remote sensing data sources, including MODIS, Landsat, and DEM data, to improve the regression precision of population mapping. The daytime remote sensing data provide supplementary residential information, such as community distribution, building level, and area. The information helps in distinguishing the actual residential regions and excluding many unrelated regions producing interferential light intensity and records by NTL sensors, thereby negatively affecting population mapping. The third category is with remote sensing and GIS-based social sensing data [42,43]. Although residential details have been considered important in population mapping and are partially extracted through daytime remote sensing data, the indexes directly correlated to population distribution remain lacking. The adoption of social sensing data can provide complementary residential information. Social sensing data are considered promising in supplying information of population dwelling environment and distribution details. A few studies have adopted simple social sensing data, including point of interest (POI) and OpenStreetMap (OSM) data, to realize the functional area identification of cities and establish multiple regression models [44,45]. However, the subjective functional area division cannot fully reveal the complex human activities and distribution pattern. The lack of actual human activity information restricts the improvement of the accuracy and reliability of these methods. As the main research object, population information should be considered reasonably in the population mapping analysis, but it has been ignored by previous studies.

Social media data, which are social sensing data highly correlated with human activities and produced by web users or platforms, are currently broadly used in urban studies [46,47]. With the popularization of mobile communication equipment and development of mobile sensors, human activities can be accurately recorded through their information sharing in social media platforms, such as Facebook and Twitter. Social media data are among the most important resources in big data analysis due to their capability to provide substantial data, for example, Twitter, which can provide more than 10 billion records monthly [48].

Apart from their abundant semantic information, social media data can also record the locations of mobile users, thereby increasing their potentiality in semantic and broad spatial analyses, including commercial site selection, urban functional zoning, and population migration [49,50]. The present study used the Sina Weibo check-in data, a type of location-based (location-based service (LBS)) social media data that include people's daily life activities and current locations, to formulate the population distribution, considering the high correlation between social media data and human activities. Such data are produced by Sina Weibo platform, the largest microblog platform in China, with over 4.3 hundred million active users.

We proposed a novel model for disaggregating the census data into grid level with a resolution of 25 m by 25 m. We attempted to reveal the population distribution pattern by combining International Space Station (ISS) nighttime photography, POI, and Sina Weibo check-in data. The multiple data sources were integrated through a random forest (RF) model and then used to investigate their relationship with census population data in town scale. The trained model was then used to disaggregate census data into grid scale. The accuracy was proven to reach 91% and superior to previous studies through a comparison with WordPop data. We discussed the advantage of using our method in handling residential regions with similar light intensity and illustrated its possibility in further improving the

accuracy of population mapping when check-in data are acquired in appropriate time and volume. This study shows the great potential of social media data for use in population mapping.

## 2. Materials

### 2.1. Study Areas

Wuhan, the capital city of Hubei Province, is located in the central part of China. As one of the biggest central cities in China, Wuhan is the core city in the Yangtze River Economic Zone.

With its superior economy and education resources, Wuhan has attracted a large amount of permanent resident population with complex structure. Statistical data show that the permanent residents reached 11.1 million until 2018. The rapid population growth of Wuhan causes the increase in many challenges for the management of public health, public security, urban traffic, and environment pollution. Hence, developing strategies to guide resource allocation and city planning on the basis of the detailed population distribution information is essential.

Wuhan consists of 13 municipal districts and 199 subdistricts. Figure 1a,b shows the location and administrative division of Wuhan. The main urban areas include seven districts, as shown in Figure 1c. With the continuous development and expansion of Wuhan City, nearby economic development zones and blocks are also listed into the main urban areas. Thus, the 98 subdistricts in the block level of the seven districts along with 15 nearby subdistricts with high development level were preliminary selected as our study areas. However, 113 subdistricts were finally selected due to the missing data of three subdistricts distributed near the edge of Wuhan, as shown in Figure 1d.

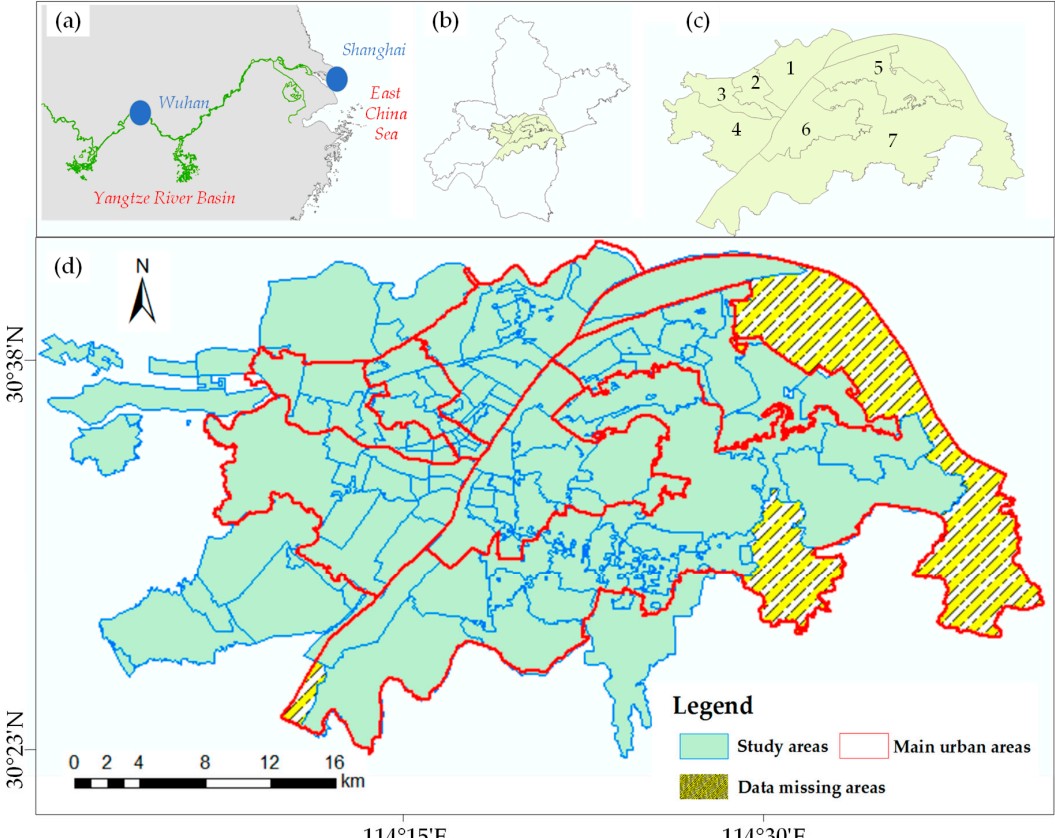

**Figure 1.** (**a**) Location of Wuhan in the Yangtze River Basin. (**b**) Administrative division of Wuhan in the municipal district level. (**c**) Seven districts in the main urban areas of Wuhan (from one to seven): Jingan, Jianghan, Qiaokou, Hanyang, Qingshan, Wuchang, and Hongshan. (**d**) 113 subdistricts selected as our study areas.

*2.2. Data Sources*

The NTL data adopted in this study included the ISS photography products when the space station passed over China on April 6, 2011, 22:37 UTC, and covered an area of 28 km by 40 km. Although the ISS photography products were available from 1987 to now (2019), the latest census data was just provided in 2010, due to the census survey being conducted every ten years in China, which is the main consideration of choosing the photography obtained in 2011 rather than another time. The photographs were obtained using a 12.1-megapixel Nikon D3S digital still camera, the sensor of which produced red, green, and blue (RGB) photographs archived as a Nikon electric film (NEF) file.

ISS photography can provide detailed lighting information of different light sources with high resolution (25 m) on the basis of its scotopic and photopic bands [51,52]. Thus, the texture information and spectral information were considerably more abundant than those of other NTL data sources. However, the ISS images were captured at an oblique angle, and no georeference information was found in the ISS original images. Hence, image registration was conducted with Google Earth as the reference map and the resolution of ISS NIT image was kept unchanged (25 m). Moreover, 25 registration points were selected correspondingly in ISS image and Google Earth through visual observation. The result was proven accurate, with an average registration error of 2.7 m, through the overlay with Google Earth and comparison of 30 randomly selected checkpoints.

Landsat 7 Enhanced Thematic Mapper Plus (ETM+) images were used to estimate Normalized Difference Vegetation Index (NDVI). Four images covering Wuhan City under clear weather conditions in 2011 were downloaded from the website of United States Geological Survey (USGS) (https://earthexplorer.usgs.gov/). The resolution of derived NDVI image was set as 25 m through resample.

The social media data used included the location-based check-in data from Sina Weibo (an application of Sina Corporation), the largest blog platform in China. Similar to Facebook and Twitter, people share their daily life activities with words, pictures, and videos along with their current locations in the blog platform, which is called "check-in" in the Sina Weibo. Thus, a check-in record includes basic information, such as user ID, gender, age, and check-in time, and abundant semantic information, including name of check-in place, comments, and pictures. We acquired the check-in data with Sina API from 1 January 2010 to 31 December 2010, in Wuhan. The original dataset included approximately 2,020,000 records. After data cleaning, duplicate and incomplete records were eliminated, and 1,912,181 effective check-in records remained.

The POI data used were derived from the Wuhan Geographic Information Bureau. The records without complete name or category information were removed. Thus, 191,175 effective records were retained and adopted in this study. POIs were distinguished into 16 categories in accordance with their functions: Catering facilities, auto service facilities, sports facilities, residential quarters, shopping malls, life service facilities, medical facilities, hotel facilities, attractions, government agencies, cultural facilities, traffic stations, financial facilities, landmarks, factories, and communal facilities.

Road networks and building profiles were derived from OpenStreetMap data (www.openstreetmap.org) in July, 2010. As an open community formed by millions of participants, the OpenStreetMap aims to provide free map data to the public with high accuracy and timeliness. The data were updated and maintained daily by over 1.5 million map editors through aerial images and high-precision GPS data.

The population data of the 113 subdistricts were collected from the Sixth National Population Census conducted in 2010 launched by the Chinese government (http://data.stats.gov.cn/). WorldPop datasets (http://www.worldpop.org.uk/), which are the products provided by the WorldPop projects with the aim to provide open-source population datasets of Africa, Asia, and America, were used as the validation data in accuracy assessments.

The administrative boundaries of Wuhan were downloaded from the website of the National Geomatics Center of China. The data were produced in 2004. The data were directly used in the analysis because the administrative division of Wuhan between 2004 and 2010 did not change remarkably.

All spatial references of these data were unified to WGS-84.

## 3. Methods

### 3.1. Flowcharts

Figure 2 shows the flowchart of the method, which can be divided into four steps: (1) The ISS image was preprocessed, including hue saturation lightness (HSL) transformation and saturation calibration. (2) Similarity (e.g., semantic and distance) matching of check-in and POI data was performed. The main information of the matched data was then extracted via principal component analysis (PCA) and set as the input of RF regression model along with road density and light intensity. (3) Population mapping in grid scale was conducted on the basis of RF and accuracy assessment.

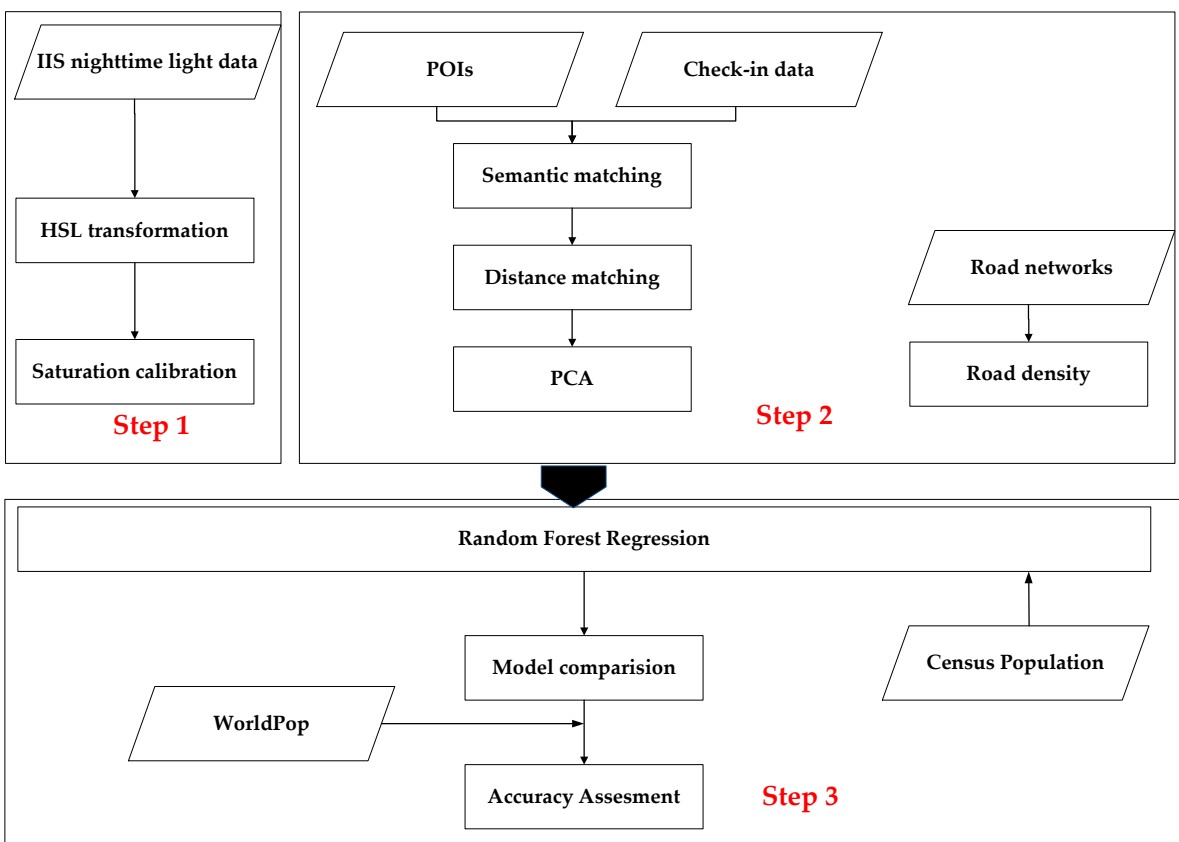

**Figure 2.** Flowchart of the proposed population mapping.

### 3.2. Preprocessing of ISS Photography

HSL transformation was performed on the original RGB ISS photography. To utilize the spectrum information of ISS data fully, the average lightness of the three bands was set as the HSL value for each pixel:

$$\text{ISS HSL}_{i,j} = \frac{1}{3} \sum \text{ISS HSL}_{i,j}(\text{R}, \text{G}, \text{B}) \tag{1}$$

where $i$ and $j$ represent the row and column of the ISS image, respectively. Thus, the HSL image was transformed from RGB bands, and the adjusted digital number (DN) values were distributed in a relatively small range (0–255) but broader than the DMSP/OLS NTL products (0–63). Hence, the saturation phenomenon still existed in the core urban areas. To reduce the saturation influence, we utilized the Vegetation Adjusted NTL Urban Index (VANUI), which is often used in processing DMSP/OLS data. and the spatial resolution of ISS NTL image was set as 25 m [53,54]:

$$\text{ISS NTL}_{i,j} = \left(1 - \text{NDVI}_{i,j}\right) \times \text{ISS HSL}_{i,j} \tag{2}$$

### 3.3. Similarity Matching of Mobile Check-In Data and POI Data

The Weibo check-in data, as a type of LBS data, describes the general distribution pattern of Weibo users. Their quantity also reflects the activity degree of Weibo users in different regions. An important characteristic of Weibo check-in data is that they are recorded on the basis of the location of POI, indicating that the check-in action occurs near a specific facility [55]. Thus, we can infer the popularity of a POI on the basis of the number of Weibo check-in data distributed around it. The POI and Weibo check-in data are two types of social sensing data and are produced by two different platforms: Wuhan Geographic Information Bureau and Sina Corporation, respectively. Hence, the element expressions of these data sources are different due to their varied data producing standards and mapping methods, leading to the deviation of facilities in names and locations. To realize the accurate matching of the two data sources, we utilized their semantic and location information and proposed a spatial data matching process. Considering that the semantic information was often more accurate and reliable than location information and that POI data were often concentrated, especially in hot spots, the processes were divided into two steps: (1) semantic similarity matching and (2) distance similarity matching.

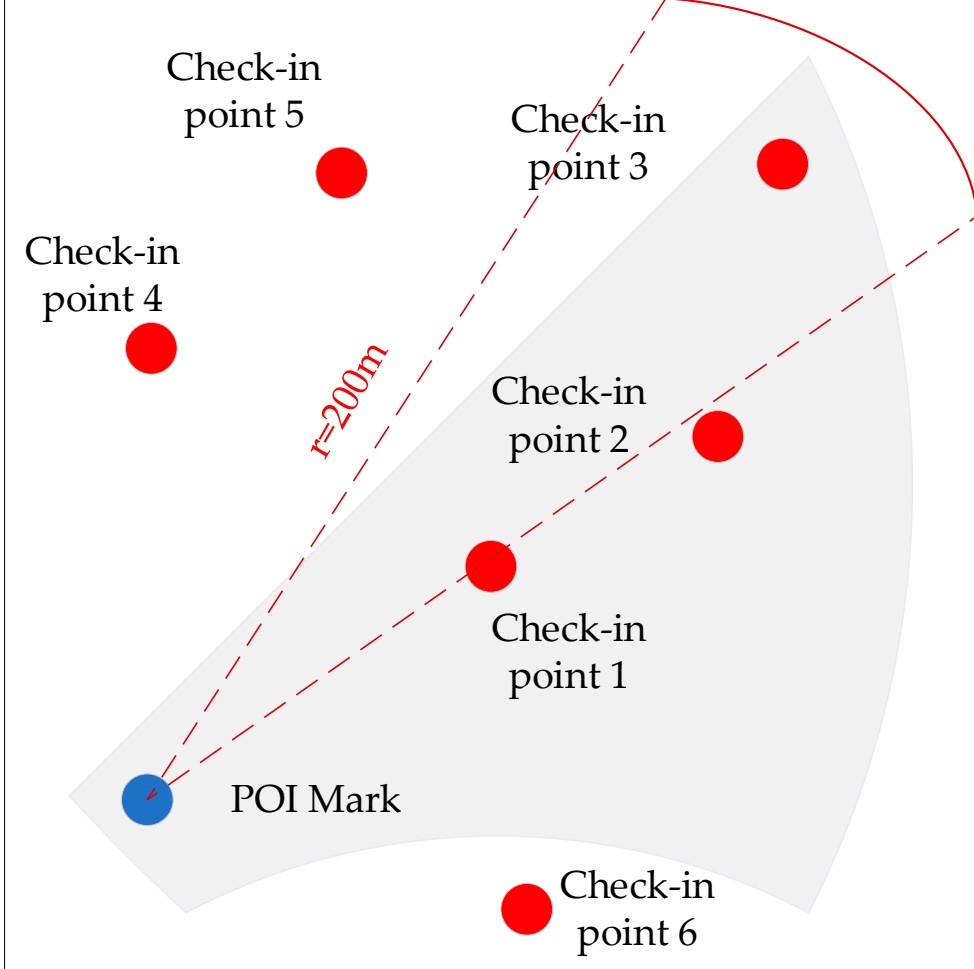

**Figure 3.** Example of a search range with 200 m radius. Red points denote the check-in points, the blue point represents the point of interest (POI) mark of a facility, and gray area denotes the outline of the facility. When the radius was set to 200 m, all check-in points belonging to the facility (Check-points 1, 2, and 3) were included in the search range.

Prior to similarity matching, a search range was formed around each POI to find check-in points with the same location name. The number of check-in points that matched the POI was recorded as the popularity evaluation index of the POI data. The search range was determined on the basis of

the maximum coverage area of a facility in the POI data source, the maximum radius of which was 187 m. Hence, the radius of our search range was set to 200 m, which is larger than the maximum coverage area, to ensure that all objects with the same location name can be found accurately, as shown in Figure 3. Another reason for setting the radius to 200 m is to avoid the confusion of some facilities with numerous subbranches distributed far from each other with the same name.

### 3.3.1. Semantic Similarity Matching

The names of the same facility in the two data sources were not always the same due to the different mapping expressions. The differences may be caused by various reasons, such as word omission, transposition, and abbreviation [56,57]. To solve this problem, we adopted a data matching method based on a string similarity evaluation method to determine the facilities with similar names [58]. Initially, the name of each facility was normalized into a token set, for example, the names M1 = "Hubei Provincial Museum" and M2 = "Museum of Hubei Province" were transformed into tokens that include three words, that is, T1 = {"Hubei," "Provincial," and "Museum"} and T2 = {"Museum," "Hubei," and "Province"}, respectively. The string similarity evaluation method was then used to resolve the task. The string similarity was calculated in accordance with the minimum number of characters, including deletions, additions, and substitutions, required to realize the transformation, namely, Damerau–Levenshtein distance [59,60]:

$$S_{p,q} = 1 - \frac{DamLev(p,q)}{Max(Length(p), Length(q))} \tag{3}$$

where $DamLev(p,q)$ denotes the Damerau–Levenshtein distance between words $p$ and $q$ in the two tokens. $Length(p)$ and $Length(q)$ denote the character lengths of words $p$ and $q$, respectively. S denotes the similarity index, ranging between 0 and one. Each word in token T1 is compared with the words in token T1. Thus, a matrix that consists of $S_{p,q}$ is formed. The largest value of each column denotes the best match between words and is recorded as $S_{optimal}$. The overall similarity of tokens T1 and T2 is then summed as follows:

$$O_{T1,T2} = \frac{\sum_{k=1}^{l_{max}} S_{optimal}(k)}{\mu} \tag{4}$$

where $O_{T1,T2}$ denotes the overall similarity between tokens T1 and T2, $l_{max}$ denotes the size of the large token set, and $\mu$ denotes the mean length of the sets of tokens derived from the two strings. Thus, the similarity of M1 and M2 was 0.987 based on the method. To distinguish the identification threshold value of semantic matching, we artificially selected 5000 pairs of name expressions of the same facilities (located in the same building profile) from POI and check-in data on the basis of their locations. The minimum value (0.756) was set as the identification threshold value. When the overall similarity between the check-in points and target POI is larger than 0.756, these check-in points match the POI. On the contrary, when the overall similarity is smaller than 0.756, the check-in points are considered unrelated with the target POI.

### 3.3.2. Distance Similarity Matching

Through semantic similarity matching, approximately 83% Weibo check-in data were successfully matched with the POI data. The 17% data remaining were transported to their nearest POI on the basis of a distance similarity model. According to *Tobler* [61], near things are more similar to each other than distant things. Thus, the popularity of a POI can be reflected by the check-in data around it. We adopted a nearest neighbor method [62,63]:

$$D_{(a,b)} = 1 - \frac{distance(a,b)}{min(distance(a,b_2), distance(a_2,b))} \tag{5}$$

where *a* denotes a POI point, *b* denotes check-in data, $a_2$ denotes the POI second nearest to check-in point *b*, and $b_2$ denotes the check-in point second nearest to POI point *a*. $D_{(a,b)}$ denotes the distance similarity between *a* and *b*. To ensure that the matching errors between semantic matching and distance matching are small and homogeneous, the distance similarity threshold value was set in accordance with the average distance similarity of the data in semantic matching at −2.75. Thus, the check-in data with a distance similarity larger than −2.75 will match the target POI point.

After semantic and distance matching, 99.1% of the records successfully matched the POI data, except for 0.9% of the records, which were mainly distributed far from the city center. The unmatching may be caused by the different update frequencies between data sources. The check-in data were updated daily, but the POI data were updated monthly or seasonally. Thus, some new facilities may not be found and recorded by the municipal department in the POI dataset, especially when they are distributed far from the city center.

### 3.4. Principal Component Analysis of Point of Intrests Data

Fourteen categories of POI data with different social service functions were considered, leading to different correlations with population distribution. Therefore, simply summing these POI points is inappropriate. In the previous study, the spearman correlation was adopted to distinguish the relationship between each POI category and the population census data [26]. The correlation coefficient was then set as the weight of each POI category. However, the method ignored the correlation among these POI categories, which has been discussed in previous studies [64]. The basic requirement of RF is to ensure the independence between input variables. Thus, PCA was conducted to eliminate the dependence between variables. The components that accumulated 85% were selected as the inputs of RF [64,65].

We then established a popularity evaluation index to describe the popularity of each component by using the matched check-in data:

$$\text{Component}_i(x) = \sum_{j=1}^{j=14} a_{ij} \times Nor.\big(\text{POI}_j(x)\big)) \tag{6}$$

$$\text{P}_i(x) = \text{Component}_i(x) \times \sum_{j=1}^{j=14} a_{ij} \times Nor.\big(\text{CK}_j(x)\big) \tag{7}$$

where $\text{Component}_i$ represents the *i*th component of the PCA results, *x* represents a geographic unit, $a_{ij}$ represents the weight of the *j*th category of POIs in the *i*th component, and $Nor.\big(\text{POI}_j(x)\big)$ represents the normalized number of the *j*th POIs in unit *x*. $\text{P}_i(x)$ represents the popularity evaluation index of the *i*th component in unit *x*, and $Nor.\big(\text{CK}_j(x)\big)$ represents the normalized number of check-in points that matched the *j*th POIs.

### 3.5. Population Mapping with RF Model

The complex pattern cannot be described simply by using linear regression models due to the spatial heterogeneity in population distribution. Thus, we adopted the RF method, which is a classical nonlinear regression model in machine learning [44,66,67]. As a representative approach of ensemble learning, RF consists of a group of mutually independent decision trees based on classification and regression three [68,69]. Each tree is trained with partial data and provides a prediction result. The result of RF will depend on the quantity of the same results provided by these trees [70]. The main steps of RF in population disaggregation are described as follows [71]:

Preparation and training: Three categories were set as the inputs of RF, namely, the popularity of POIs (described by *P* values), light intensity, and road density (described by the ratio between road

length and area in a unit). These inputs were considered in the subdistrict level, the same as the scale of census data, which was set as the output of the RF.

Samples and growth: On the basis of the bootstrap method, $n$ cases of the training samples were extracted from the original dataset to train the decision trees. Moreover, 30% of the original was selected out of the bag as the validation set. The regression tree with $n$ nodes was built initially, and then the value of $n$ was finally confirmed in accordance with the feature bagging of the trees.

Validation and prediction: After the construction of regression trees, the pixel level population distribution was predicted through the trained RF. The performance was then evaluated through the validation set [72]:

$$R^2 = 1 - \frac{\sum_{i=1}^{n}(\hat{y}_i - y_i)^2}{\sum_{i=1}^{n}\left(y_i - \overline{y}_i\right)^2} \tag{8}$$

where $\hat{y}_i$ denotes the population of pixel $i$ derived from WordPop data, $y_i$ denotes the estimated population data through RF, $\overline{y}_i$ denotes the average value of the estimated data, and $n$ denotes the number of pixels.

## 4. Results

### 4.1. HSL and VANUI Calibration Results of ISS Photography

Figure 4 presents the effects of HSL and VANUI calibration. Figure 4(a1) shows the original ISS RGB image after registration, where the city outline is described dispersedly in three bands. Figure 4(a2) exhibits the results after HSL processing, where the information of three bands is integrated. The results indicated that after HSL processing, a clear city structure was maintained. Figure 4(a3) shows the results after VANUI calibration. Further striking contrast between regions with different DN values was shown in the VANUI image.

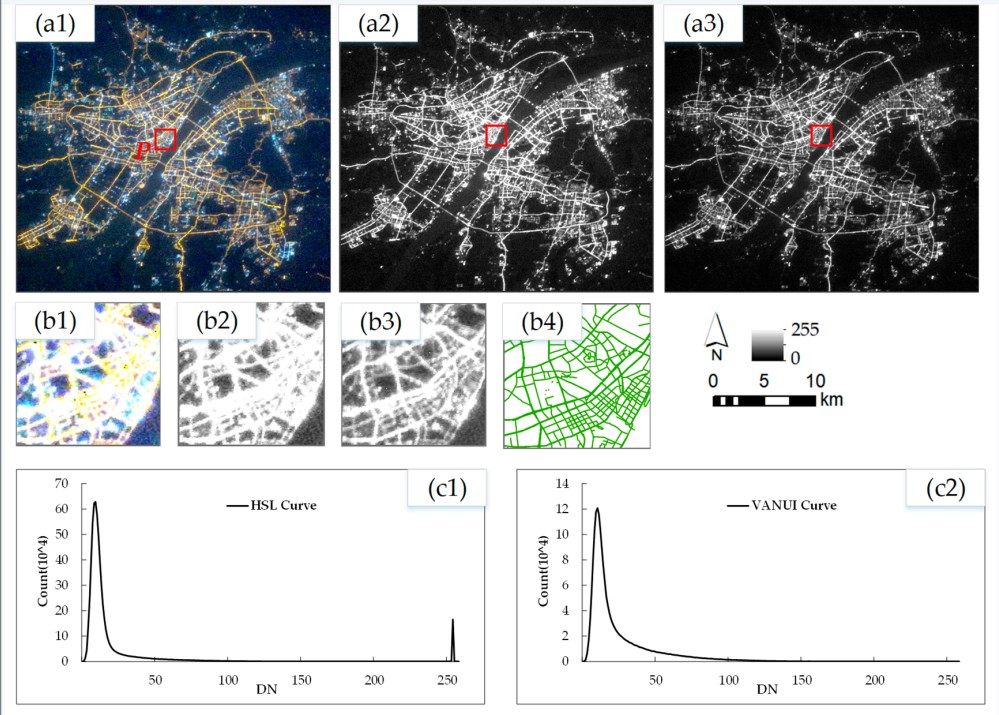

**Figure 4.** (**a1**–**a3**) Comparison of original International Space Station (ISS), hue saturation lightness (HSL), and Vegetation Adjusted NTL Urban Index (VANUI) images. (**b1**–**b4**) Sample region showing the comparison of original ISS, HSL, VANUI images and actual road networks. (**c1**,**c2**) Comparison of DN curve between HSL and VANUI images.

To capture further detailed differences among three images, we selected a small region in the core area of Wuhan (labeled with *P*). Figure 4(b1–b3) show the original, HSL, and VANUI images of *P*, respectively. Although the image information was improved in the HSL image, its image contrast was relatively low and the brightness of most road pixels was high. The brightness contrast between road pixels was improved in the VANUI image, thereby clearly distinguishing the main and feeder roads close to the actual OSM road network, as shown in Figure 4(b4). The quantitative differences between HSL and VANUI are further described in Figure 4(c1,c2) through the DN curve. The comparison indicated that the quantity of pixels with high DN values was reduced through VANUI processing and the supersaturation phenomenon was eliminated as the high DN values in the HSL image disappeared in the VANUI image.

### 4.2. Results of Similarity Matching

Figure 5 shows the similarity matching between social media and POI data, where a sample region was selected to show the matching process. The sample region was located in the northeast of Wuhan, one of the city centers.

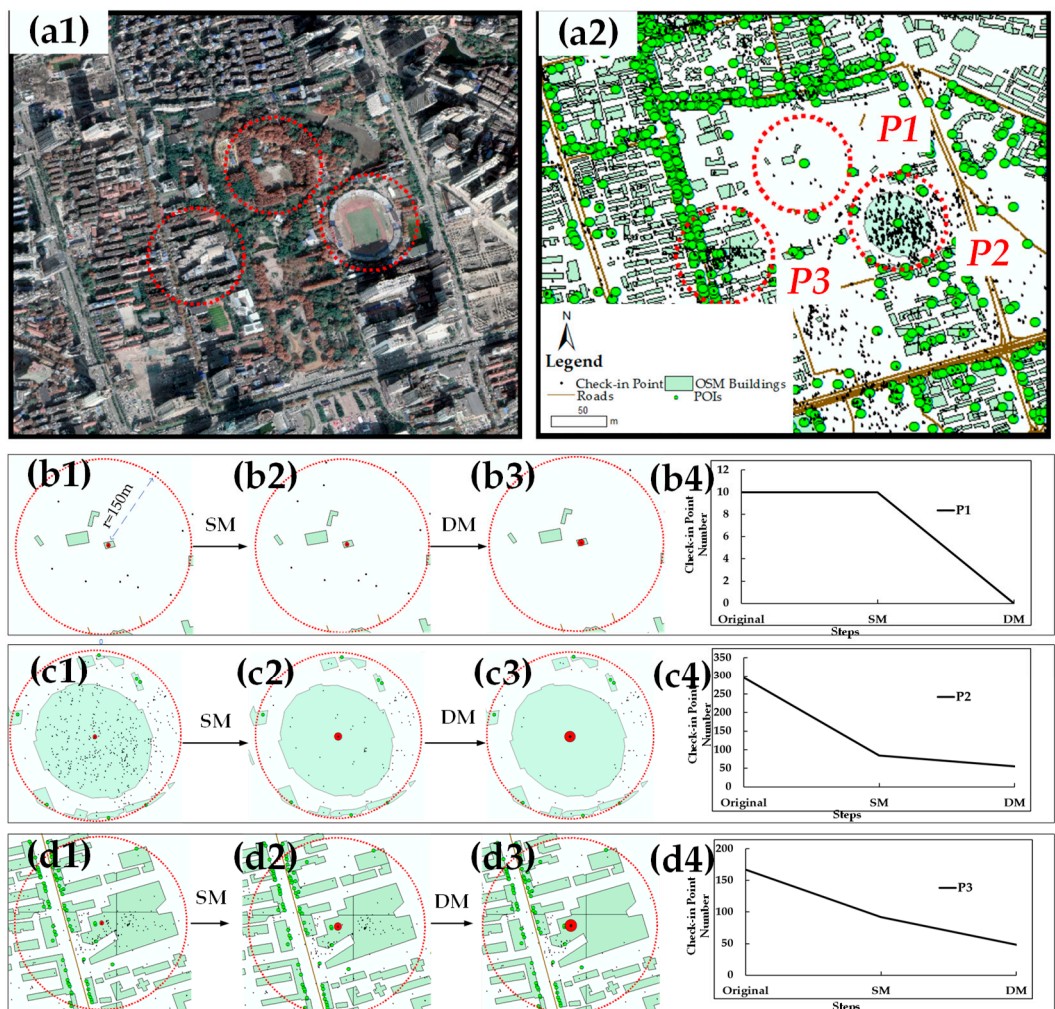

Note: SM (semantic matching), DM (distance matching).

**Figure 5.** (**a1**,**a2**) Selected three sample regions: P1, P2, and P3. (**b1**–**b4**) Similarity matching of P1. (**c1**–**c4**) Similarity matching of P2. (**d1**–**d4**) Similarity matching of P3.

Figure 5(a1) shows the Google Earth image of the sample region, and Figure 5(a2) presents the vector map of the sample regions. The distribution of POIs, check-in points, buildings extracted from

OSM, and roads was shown in the vector map. To describe the matching process in detail, three representative POIs were selected, as described by P1, P2, and P3. The search range of the three POIs was represented by red circles with the radius of 200 m.

(1) P1 is a building with small area, the search range of which only contained the single POI and several dispersive check-in points. In comparison with the original dataset shown in Figure 5(b1), the number of check-in points was unchanged after semantic matching, as shown in Figure 5(b2), indicating that the check-in activities did not occur in P1. However, all check-in data were aggregated to P1 after distance matching, indicating that P1 was the nearest POI around the check-in points. P1 could be a representative of places, such as public facilities and industrial regions, distributed far from the city centers. Thus, few human activities occurred at these POIs; as a result, semantic matching had difficulty aggregating the check-in points into these POIs. Therefore, distance matching could be the most important approach for solving the problem and proved to be effective in aggregating all check-in points into the sample POI, as shown in Figure 5(b3). The line chart in Figure 5(b4) shows the change trend of the check-in data in the two similarity matching procedures.

(2) P2 is a sports center with large area, the searching range of which contains several other POIs and hundreds of check-in points. In comparison with the original dataset in Figure 5(c1), most check-in points were aggravated in the semantic matching procedure, as shown in Figure 5(c2), and only a few check-in points were aggregated in the distance matching procedure, as shown in Figure 5(c3). The line chart in Figure 5(c4) clearly shows the change trend. P2 could be a representative of places with large areas and relatively high popularity compared with nearby POIs, such as large residential areas, parks, famous scenic spot, and sports centers. The visiting frequency of these POIs was distinctly higher than those of other nearby POIs. Hence, the location names of the check-in points in these POIs could be easily recorded, causing the high ratio of semantic matching.

(3) P3 is a restaurant in a shopping mall, the searching range of which contains dozens of other POIs and hundreds of check-in points. In comparison with the original dataset in Figure 5(d1), nearly half of check-in points belonged to the POI through semantic matching in Figure 5(d2), and the others belonged to the POI through distance matching in Figure 5(d3). The line chart in Figure 5(d4) describes the quantitative change trend. P3 could be a representative of commercial places with numerous facilities with similar scales and functions. Many interference factors existed in these regions, making the semantic and distance matching effective in different places. Therefore, the combination of the two matching methods could be the best approach in these regions.

The similarity matching was conducted to all POIs of Wuhan. The check-in points of the 14 categories of POIs were aggregated. Table 1 shows the results.

**Table 1.** Overall similarity matching results.

| Matched Check-In Points | POIs | Rate (%) | Top 3 Maximum POI Categories | Average |
|---|---|---|---|---|
| ≥20 | 7486 | 3.901 | Shopping malls, residential quarters, catering facilities | 33 |
| ≥10 and <20 | 69,532 | 36.261 | Shopping malls, residential quarters, attractions | 15 |
| ≥1 and <10 | 103,425 | 53.926 | Catering facilities, government agencies, hotel facilities | 6 |
| 0 | 11,342 | 5.912 | Auto service facilities, cultural facilities, factories | 0 |

We counted the check-in points that matched each category of POIs. Table 2 shows the results.

**Table 2.** Similarity matching results of each POI category.

| Category | POI Number | Number of Check-In Points | Average Check-In Points | Maximum Number of Check-In Points |
|---|---|---|---|---|
| Catering facilities | 43,909 | 491,294 | 11.19 | 242 |
| Auto service facilities | 539 | 172 | 0.32 | 14 |
| Sports facilities | 3459 | 6185 | 1.79 | 21 |
| Residential quarters | 41,041 | 493,514 | 12.02 | 395 |
| Shopping malls | 43,093 | 480,311.9254 | 11.15 | 199 |
| Life service facilities | 8304 | 67,162 | 8.09 | 87 |
| Medical facilities | 3872 | 40,720 | 10.52 | 174 |
| Hotel facilities | 3551 | 83,578 | 23.54 | 89 |
| Attractions | 4825 | 87,523 | 18.14 | 105 |
| Government agencies | 9725 | 113,436 | 11.66 | 95 |
| Cultural facilities | 4486 | | 0.00 | 74 |
| Traffic stations | 8474 | 29,683 | 3.50 | 23 |
| Financial facilities | 1626 | 1592 | 0.98 | 12 |
| Landmarks | 2875 | 3803 | 1.32 | 25 |
| Factories | 2421 | 2418 | 1.00 | 14 |
| Communal facilities | 8975 | 10,789 | 1.20 | 31 |
| Sum | 191,175 | 1,912,181 | 10.00 | |

*4.3. Results of Principal Component Analysis*

Table 3 shows PCA results. With reference to a previous study [62], the threshold value was set to 85%. Thus, the top four components were set as the new variables of RF along with the NTL and road density.

**Table 3.** Variance and cumulative accuracy of each component.

| Component | Variance (%) | Cumulative (%) | Component | Variance (%) | Cumulative (%) |
|---|---|---|---|---|---|
| 1 | 34.275 | 30.275 | 9 | 1.032 | 98.143 |
| 2 | 23.212 | 57.487 | 10 | 0.712 | 98.855 |
| 3 | 17.711 | 75.198 | 11 | 0.57 | 99.425 |
| 4 | 10.658 | 85.856 | 12 | 0.207 | 99.632 |
| 5 | 4.226 | 90.082 | 13 | 0.175 | 99.807 |
| 6 | 3.029 | 93.111 | 14 | 0.126 | 99.933 |
| 7 | 2.075 | 95.186 | 15 | 0.058 | 99.991 |
| 8 | 1.925 | 97.111 | 16 | 0.009 | 100 |

As shown in Table 3, Components 1–4 described 34.275%, 23.212%, 17.711%, and 10.658% information of the 16 variables, respectively. The results indicated that Components 1–4 described 85.856% the total information of input, which was more than the threshold value (85%). Thus, the top four components of PCA were set as the inputs of the RF. Table 4 shows the relationship between Components 1–4 and the 16 categories of POIs.

**Table 4.** Relationship between Components 1–4 and the 16 categories of POIs.

| Variable | Component 1 | Component 2 | Component 3 | Component 4 |
|---|---|---|---|---|
| Catering facilities | 0.558 | 0.081 | 0.074 | −0.101 |
| Auto service facilities | 0.002 | −0.177 | 0.156 | 0.004 |
| Sports facilities | 0.082 | 0.101 | −0.112 | 0.063 |
| Residential quarters | 0.126 | 0.492 | 0.073 | −0.172 |
| Shopping malls | 0.494 | 0.079 | −0.276 | 0.024 |
| Life service facilities | 0.429 | 0.319 | 0.047 | −0.024 |
| Medical facilities | 0.137 | 0.142 | 0.291 | 0.095 |
| Hotel facilities | −0.062 | 0.004 | 0.067 | 0.249 |
| Communal facilities | 0.042 | −0.248 | 0.295 | 0.044 |
| Attractions | 0.036 | −0.164 | 0.135 | 0.029 |
| Government agencies | −0.294 | 0.041 | 0.094 | 0.117 |
| Cultural facilities | 0.056 | 0.108 | 0.071 | 0.071 |
| Traffic stations | 0.204 | 0.095 | −0.097 | 0.103 |
| Financial facilities | 0.075 | 0.008 | 0.108 | 0.096 |
| Landmarks | −0.206 | −0.285 | 0.134 | 0.059 |
| Factories | 0.007 | −0.102 | 0.085 | −0.198 |

Table 4 shows that Component 1 has a high positive relationship with catering facilities (0.558), shopping malls (0.494), and life service facilities (0.429). However, Component 1 shows a negative relationship with government agencies (−0.294) and landmarks. Thus, this component mainly describes the commercial information. Component 2 shows a high positive relationship with residential quarters (0.492) and life service facilities (0.319). Thus, Component 2 mainly describes the residential information. Component 3 shows a high positive relationship with medical facilities (0.291) and communal facilities (0.295). However, Component 3 shows a negative relationship with shopping malls (−0.276). Thus, this component mainly describes public service information. Component 4 mainly describes hotel information (0.249).

### 4.4. Results of Population Mapping

Figure 6 shows the results of estimated population mapping at a fine resolution of 25 m by 25 m, where the regions with deep colors indicate a high population density. A remarkable difference existed between core urban and rural areas. A high population density (more than 40 people per unit grid) was found in most subdistricts of Wuchang, Jiangan, and Hongshan, where hundreds of universities were occupied by millions of students. Highly correlated with the city urbanization procedure, the moderate level population density was found near city centers, including the main parts of Qiaokou and Jiangan Districts with several large lakes and mountains and less inhabitant or residential regions. A low population density (lower than 10 person per unit grid) was found in regions far from city centers. The results indicated that the population distribution highly correlated with city development. The population was likely to integrate in regions with high convenience, improved resource allocations, and good environment.

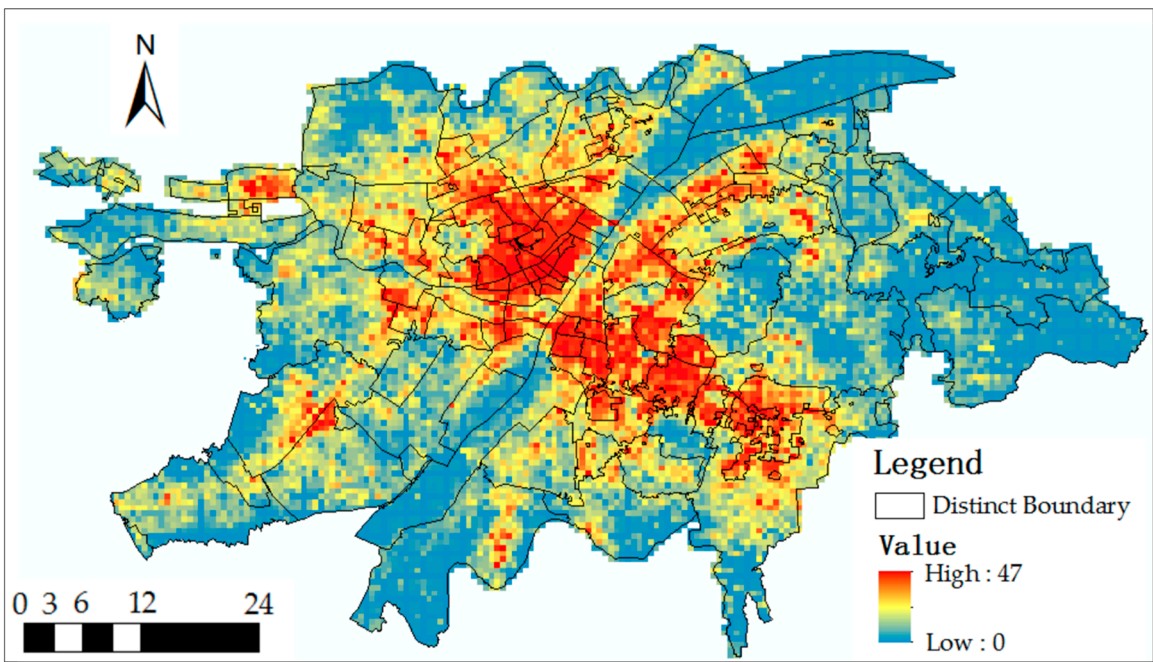

**Figure 6.** Population mapping results of Wuhan City.

To verify the accuracy of the results, the WordPop data of Wuhan in 2010 were adopted in this study. A neighborhood resampling algorithm with a four by four filter was applied considering that the spatial resolution of WordPop data and our results was different (WordPop, 100 m; our results, 25 m). We evaluated the accuracy in two aspects, namely, error and error percentage, because the population density was different between regions. Figure 7 indicates the errors between WordPop and our estimated results. The statistic results indicate that the errors were mainly distributed between 0 and 47. The regions with high errors were mainly distributed near the core urban areas. In these regions, the population density was relatively high, causing higher errors than in other regions. On the contrary, the regions with low errors were mainly distributed far from the core urban areas due to the small quantity of population in these regions.

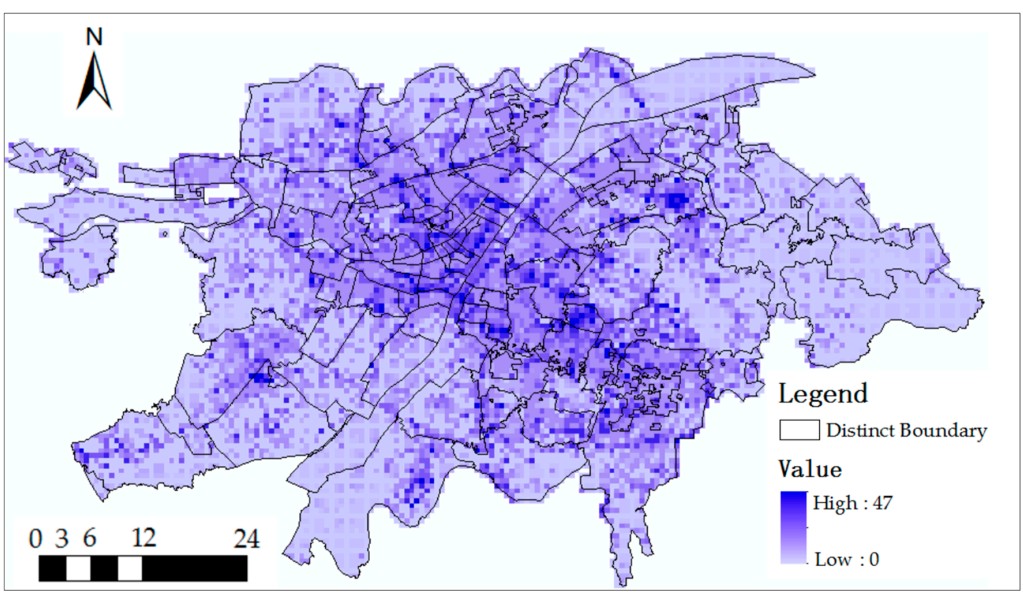

**Figure 7.** Errors of estimated population mapping results.

To evaluate the model accuracy further objectively among different regions, the error percentage was calculated and shown in Figure 8.

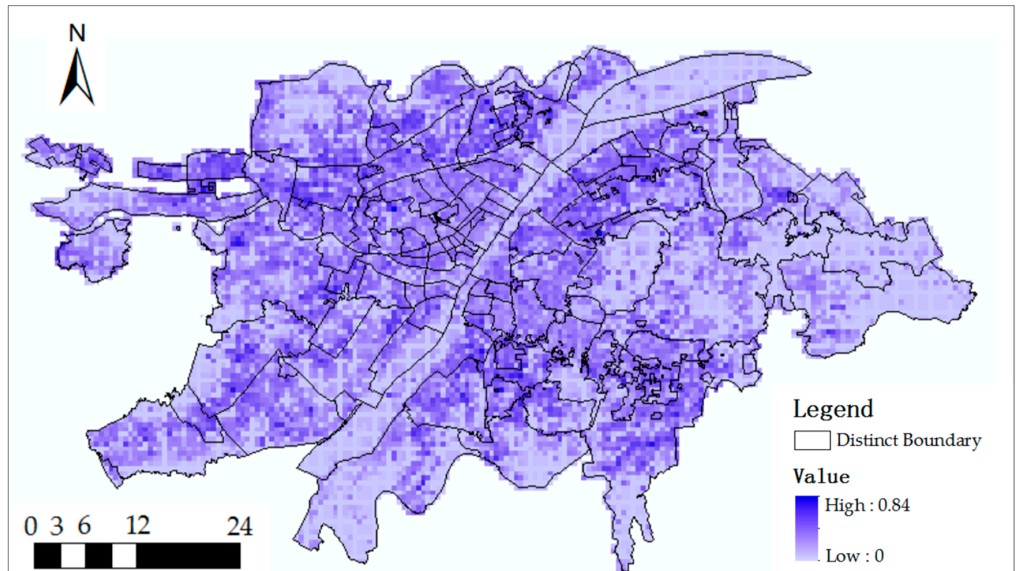

**Figure 8.** Ratio of estimation errors.

We can find the error percentages were distributed between 0 and 0.84. The results indicated that the error percentages of most regions were small. The error percentages of the middle and eastern regions of Wuhan were relatively smaller than those of the west and southern regions, indicating that the estimation accuracy in city centers was higher than that in subcenter regions. The phenomenon may be caused by the uneven distribution of POI and check-in data between city centers and subcenter regions. The insufficient inputs may restrict the estimation accuracy in these subcenter regions.

*4.5. Accuracy Assesment*

To investigate the influence of check-in data on improving the population mapping accuracy, we compared the two population mapping results before and after adopting the location-based check-in data. Figure 9 shows the comparative results, where the splashes describe the correlated relationship between the estimated population and WorldPop data in the grid scale. For comparison, the two datasets are resampled into a uniform spatial resolution (100 m by 100 m). Figure 9a shows the correlation between the Wordpop and population mapping results without considering the check-in data. Figure 9b presents the correlation between the WordPop and estimated results considering the check-in data. The population quantity in each grid may cover a large range, from 0 to nearly 1000 because the urbanization is considerably different among various parts of a city. Thus, separately evaluating the fitting precision of different regions is essential. The regions can be distinguished into three categories, namely, high, moderate, and low, in accordance with their population quantity. Regions with high population density contained the top 20% population. Conversely, regions with low population density contained the bottom 20% population, and those with moderate population density contained the other 60% population. The high, moderate, and low regions were represented by three colors: red, yellow, and green, respectively.

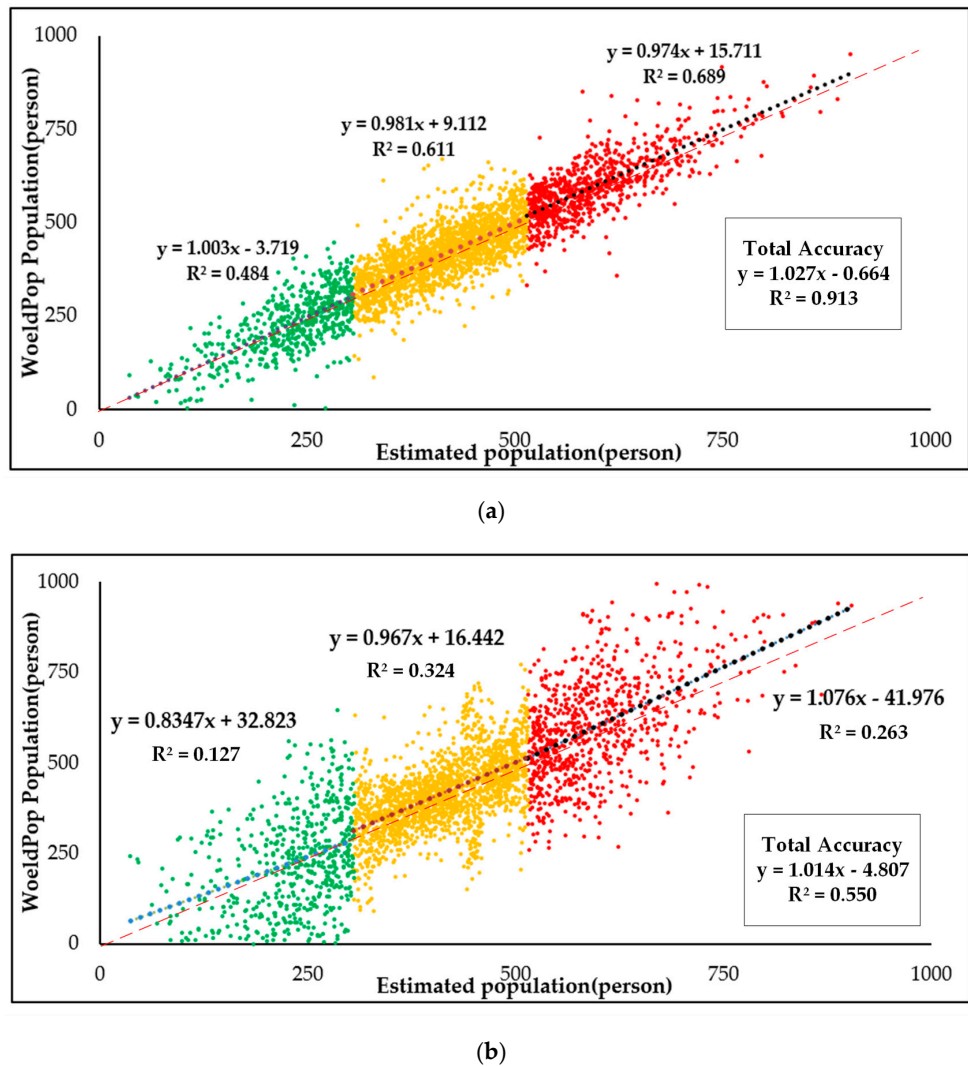

**Figure 9.** Correlation between the WordPop and population mapping results with (**a**) and without (**b**) considering the check-in points.

The results indicate that the adoption of check-in data highly improved the total estimated accuracy (from 0.550 to 0.913). The fitting curve of the three point categories indicates that the fitting accuracy of small population quantity regions was improved (from 0.127 to 0.484) after adopting check-in data.

## 5. Discussion

### 5.1. Advantages of Using Check-In Data

Similar to POI data, check-in data are social sensing data. Regions with high density of POIs have high possibility to attract additional check-in data. However, the activity information is not totally correlated with the distribution of POIs. The status of POIs, especially residential buildings, which are the main living places of the population, often plays a vital factor in deciding the residential capacity and attractive level of buildings [73]. Therefore, even for places with the same type of POIs located in similar streets, the residential capacity may have remarkable difference.

To show the phenomenon clearly and explain the necessity of adopting check-in data, we select three places located in nearby streets. Figure 10 shows the three sample regions. The light intensity of the three places was 14, and they contained only a single POI that belongs to the same residential category. Thus, the estimated population in these three places was the same (269) without considering

the check-in data. However, Google Earth images of the three places show that their construction situations were different. The POI in Figure 10a was an old community located near a busy road, which may increase the light intensity of the grid. The height of buildings in the community was low, thereby limiting their living capacity. The POI in Figure 10b was a community under construction, and no buildings were built on the ground. The POI in Figure 10c was a community with numerous high buildings and located far from the main road. Thus, the light intensity was mainly produced by the community itself. The statues of the three POIs had a remarkable difference, although they belong to the same category of POIs. Hence, concluding that the residents in the three communities are the same is inappropriate. This phenomenon decreases the estimation accuracy when only POIs were considered. Figure 10a–c show that the number of check-in points was different in the three communities. The number in place (c) is evidently larger than those in places (a) and (b). With the adoption of check-in data, the estimated population in the three places was efficiently distinguished, and the results were highly correlated with the WordPop data.

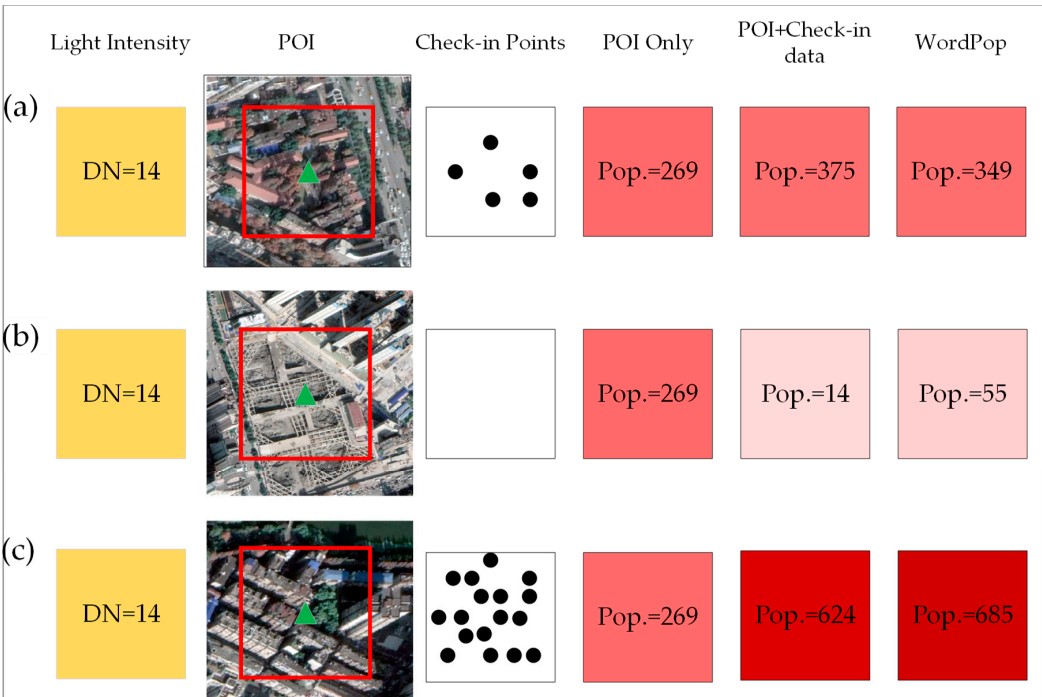

**Figure 10.** Comparison of estimated results before and after adopting the check-in data with three sample residential regions (region **a**, **b** and **c**) having the same light intensity.

## 5.2. Influence of Check-In Data Volume

As check-in data are incremental, their number may range from several thousands to over a million. The most appropriate data volume must be determined to realize the balance between computation burden and model accuracy. We extracted the check-in points randomly with an interval of 5000 from 0 to 1,910,000 and conducted the RF classification by using the selected check-in points. The curve in Figure 11 shows the estimated accuracy of the check-in data with different volumes. To reduce the error caused by the selected data, each data selection was repeated five times, and the average accuracy was set as the final accuracy of the volume.

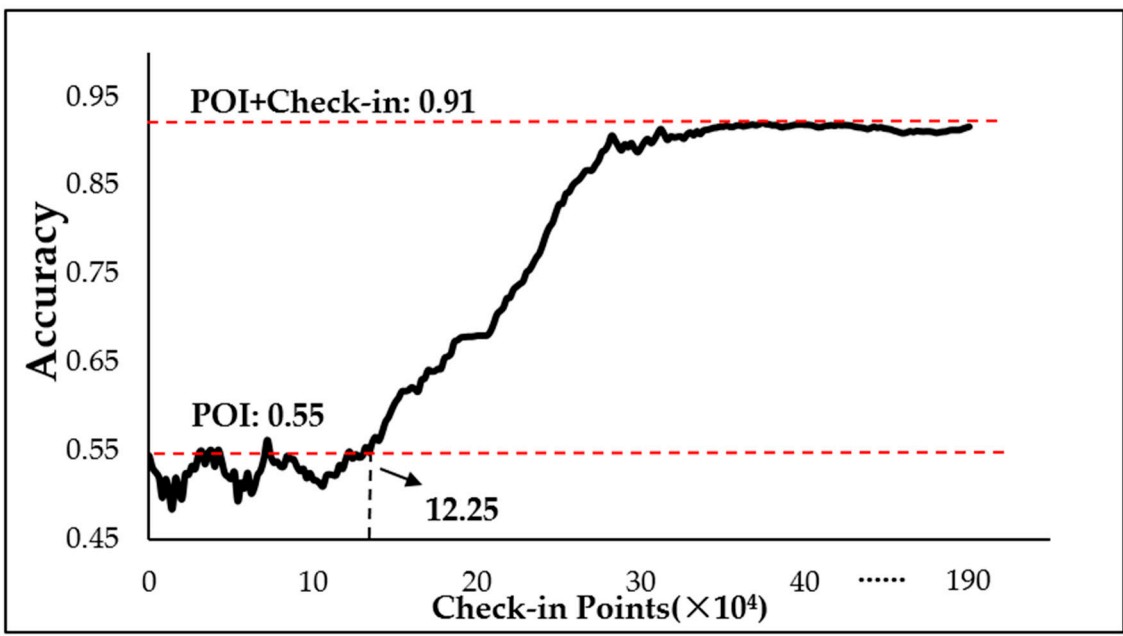

**Figure 11.** Relationship between check-in data volume and model accuracy ($R^2$).

Figure 11 shows that the estimation accuracy was simply improved with the increase in check-in data. The change trend between the model accuracy and check-in data volume can be divided into three parts: (1) from 0 to 122,500, fluctuations existed. The check-in data could only provide partial and limited information due to the small data volume, even providing negative influence to the original model without considering the check-in data. (2) from 122,500 to 300,000, the model accuracy was improved rapidly with the increase in data volume. The curve indicated that the useful information was increased with the addition of new check-in data. (3) from 300,000 to 1,910,000, the model accuracy was gradually stable at approximately 0.91. Thus, the check-in data provided sufficient information for population mapping, and the increase in new check-in data could only provide redundant information. The results declared that the volume of check-in data should be controlled in a reasonable range. The results revealed that extremely few check-in data negatively influence the original model. Correspondingly, excessive check-in data only increase the computation burden and cannot improve the model accuracy. Thus, the most appropriate data volume of check-in data for Wuhan City is approximately 300,000. The results indicate huge quantity of check-in data is not required in the estimation of population mapping. For Wuhan City, where the population is over 9,000,000(in 2010), the population distribution accuracy can be improved, compared with only NTL data, when the data volume of check-in data is larger than 122,500, which is far less the total check-in data of Wuhan. Our method is conducted based on the similarity matching between POIs data and check-in data, and the most important role for check-in data is to describe the attractive level of POIs. Therefore, the volume of required check-in data is correlated with the number of POIs. For smaller cities with less population and POIs, the estimation accuracy of population mapping can be significantly improved with the adoption of check-in data even in a small quantity. Although the most appropriate data volume of check-in data may be different in other cities due to their different economical structure and urban construction, the results indicated it is appropriate to prepare the check-in data referring to the number of POIs rather than population quantity before analysis.

### 5.3. Influence of the Acquisition Time of Check-In Data

In addition to data volume, the acquisition time is also an important factor of check-in data recorded along with the check-in behavior. The daily life of people was regular most of the time. Thus, the location of the people can be inferred [49]. For instance, most people stay in their residence in

the middle of the night and stay in their workplace during working hours. The time information in the check-in data indirectly reflects the relationship between the people and their facilities, such as temporarily shopping, permanently living, or regularly working. The relationship has been widely used to identify the city function zones and population mobility [74,75]. Hence, the adoption of check-in data acquired in different times may have different influences on the estimated accuracy. To investigate the potential influence, we divided the entire check-in dataset into four parts in accordance with the usual routine: P1 (23:00–8:00), P2 (8:00–12:00), P3 (12:00–19:00), and P4 (19:00–23:00) [13,76,77]. Figure 12 shows the time period and volume of each part.

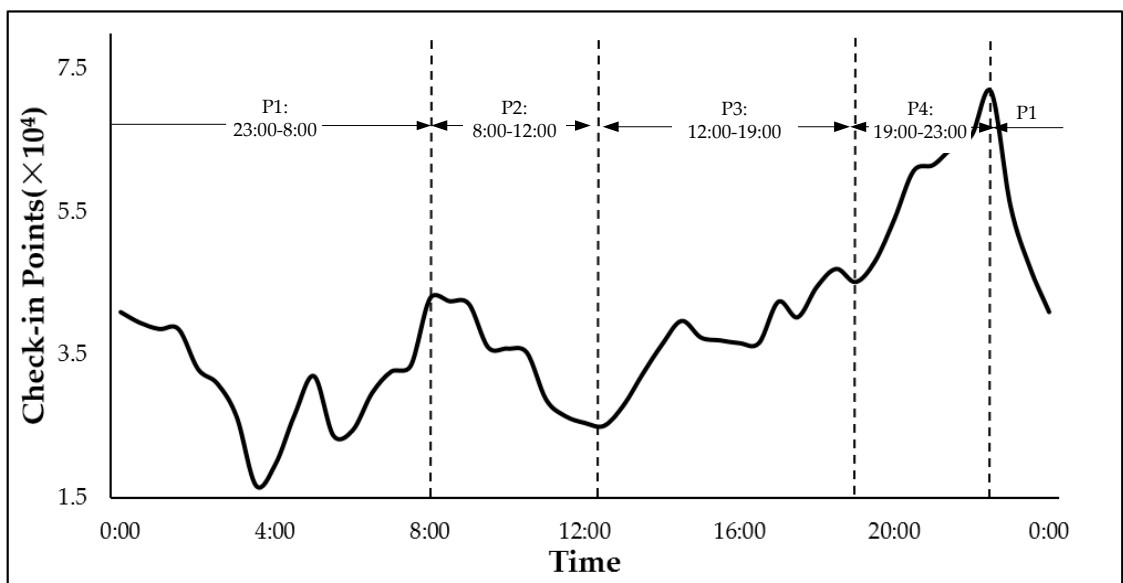

**Figure 12.** Time period and volume of four dataset components.

Figure 12 shows that the check-in activity frequently occurred during nighttime (P4: 19:00–23:00), which is the most relaxing time of the day, that is, most people have completed their work and return to their homes. The second active time period for Weibo users was at approximately 8:00, indicating that many people checks their home in the morning. A small decrease was observed in the morning (P2:8:00–12:00) and gradually increased in the afternoon (P3:12:00–19:00).

To investigate the accuracy of each component further, we separately adopted the check-in data of each part. We adopted the check-in data of four parts from 120,000 to 370,000 in an interval of 5000 because the model accuracy was relatively low when the number of check-in data was less than 122,500 and tended to stabilize when larger than 300,000. Figure 13 illustrates the accuracy curve, which shows that the total accuracy was primarily increasing with the increase in data volume until the volume reached 180,000. The accuracy of P2 tended to stabilize at approximately 0.81 when the data volume was more than 180,000. The accuracy of the other three parts, namely, P1, P3, and P4, continued to gradually increase until reaching their stable values at 0.92, 0.85, and 0.94, respectively, as shown in Figure 13. The accuracy of the four components was considerable (>0.80). The accuracy of P1 and P4 was higher than that of the other two parts, probably because most people stayed in their residences, and the check-in data had high correlations with NTL data at the time. Thus, when the data source is sufficient, adopting the check-in data provided in P1 and P4 is preferred. However, when the data are insufficient (between 120,000 and 200,000), adopting P2 check-in data is preferred.

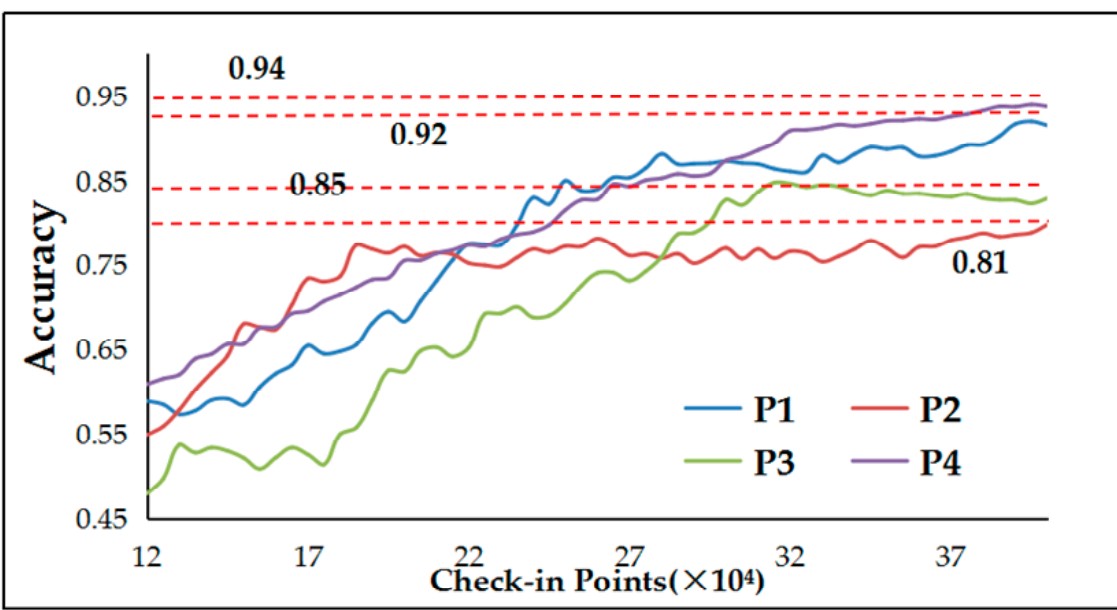

**Figure 13.** Accuracy of the model with check-in data acquired in different time periods.

During P4(19:00–23:00) and P1(23:00–8:00), the ratio of check-in data produced in residential buildings will increase, because most residents will stay at their occupancy places. Hence, the distribution of check-in data will indirectly reflect the location information of residential areas, which is lacking in NTL data, and improve the estimation accuracy of population mapping. Although may there exist differences of living habits among cities, most residents will stay at home and relax during the nighttime. Therefore, the most appropriate acquired time for check-in data is during P4 and P1, although slight differences may exist in the start or end of the time in other cities.

## 6. Conclusions

Population distribution information, as an important index in urban planning and resource allocation, has been widely studied with the adoption of remote sensing data sources in numerous studies [15,42,78]. Although multiple sources of remote sensing data have provided comprehensive spatial information of urban environment, the lack of human activity information still restricts the accuracy of population mapping. In the present study, we initially combined the location-based Sina Weibo check-in data, recorded by mobile phones, with NTL and POI data to disaggregate the census data into fine-resolution population distribution information (25 m). The statues and attractive level of POIs were quantified by establishing a two-step (semantic and distance) similarity matching model. The hybrid social sensing data information, along with light intensity derived from ISS data and road density, was placed in the RF model. These data were then used to disaggregate the census data into grid level. Through the experiment conducted in Wuhan and the comparison with WordPop data in 2010, the established model was proven to have higher accuracy (0.91) than the previous population mapping model (0.55).

To demonstrate the advantages of check-in data in population mapping, we discussed their capability in distinguishing the actual residence capacity and dwelling statue of residential buildings, which was the main problem in previous studies. As the check-in data can be easily obtained as a large volume, we also discussed the most appropriate volume for population mapping, which is 300,000 for Wuhan City. Moreover, by dividing the check-in data in accordance with the produced time, we found that the time period of check-in data can have different influences on population mapping. The check-in data produced during nighttime (19:00–23:00) and early morning (23:00–8:00) help the model achieve its highest accuracy with a reasonable data volume. This study shows the great potential of social media data for use in population mapping. In further studies, the semantic information derived

from text messages in social media data will be considered. Further detailed characteristics of the users, such as occupation, ages, and genders, will be used to establish a further accurate model. In addition, a multi-temporal analysis will be conducted with the adoption of multi-temporal NTL data and check-in data to reveal the pattern of population increasing and file the gap between two census surveys.

**Author Contributions:** L.W. and H.F. conceived and designed the main idea and experiments. L.W. and Y.W. performed the experiments. L.W. wrote the paper.

**Funding:** This research was funded by National Natural Science Foundation of China, grant number 41471323 41661086 and 91746206, National Key Research and Development Program of China, grant number 2017YFB0503500, the Science and Technology Development Project of Guizhou Province Tobacco Corporation of China National Tobacco Corporation, contract number 201407.

**Acknowledgments:** The authors would like to thank the editors and the anonymous referees for their constructive comments which have been very helpful in revising this paper. Special thanks to Xiaoyu Long for her help in data collection and support of our work.

**Conflicts of Interest:** The authors declare no conflict of interest.

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
