# Peer review of "Fine-Resolution Population Mapping from International Space Station Nighttime Photography and Multisource Social Sensing Data Based on Similarity Matching"

_remotesensing, doi:10.3390/rs11161900_

Round 1

Reviewer 1 Report

You present an interesting methodology.  You need to justify better why you even used the nightlight data.  Night lights are so dependent on affluence that it has little relevance to population. 

You incorporate some other datasets, including the mobile check-ins, that are very interesting and can't be obtained in other countries.  I would direct the paper more in this direction.

Overall, an interesting methodology and we'll written and presented.

Author Response

Point 1: You present an interesting methodology.  You need to justify better why you even used the nightlight data.  Night lights are so dependent on affluence that it has little relevance to population.

Response 1: Thanks for the reviewer’s understanding of our work and helpful comments. As the reviewer mentioned, light intensity in nighttime data is dependent on affluence, and it records not only the residential lights, but also the non-residential lights produced by commercial advertising, construction site, and security lighting. However, previous attempts have found there existed high relevance between nighttime light and city-level population, which makes NTL data become a reliable proxy for population mapping in province-level and city-level. NTL data have superiorities in describing global scale urban development at pixel level and low acquisition cost, which make them suitable for population mapping. Although NTL data cannot directly reflect population distribution in grid-scale, due to the lack of direct human activity information, the estimation rate can be significantly improved when external data sources were added in, like land use data, POIs data and social sensing data. That is also the reason why NTL data have been gradually adopted by more researchers in population mapping. We have added more words about that to make readers clear about the reason of using NTL data in this study (Line 70-75,91-94).

Reviewer 2 Report

Dear authors, I found the paper very interesting, well written and organized.

I only have some general and specific comments and a few minor issues, as per the following:

Comments:

Your analyses are mainly based on one NTL image dating back to 2011 and social media data of 2010: why did you use data of about 10 years old? Are there available more recent images and data? If yes, do a multi-temporal analysis possibly add a value to your research? Pleas provide some feedback about this.

What is the final resolution of the ISS NTL image after registration with Google Earth? Is it still 25 m? Please clarify.

Equation (2): what NDVI product have you used here? What about spatial resolution of NDVI and co-registration with NTL image? Please provide more detail about.

Figure 3: How the gray area is determined here? Please clarify.

As a more general comment, I wonder how this approach is portable/reproducible when other cities/urban environments are considered. So, the question is: how much "adaptive" is such an approach to other urban settings? How much is it dependent on data sources (in terms of quality, quantity, spatial/temporal availability, etc.)? For instance, considerations about check-in data volume and acquisition time (see figures 11, 12 and 13) how much do depend on the specific population habits of Wuhan City? Can they be significantly different as far as another city (e.g. in Europe or US) is considered?  Please provide some lines about portability issues in the discussion section.

Minor issues:

Page 10, Lines 358-359: "Figure 7" I guess should be "Figure 4" here.

Page 14, Line 434: Is "Component 3" (instead of Component 2) here?

Page 19, Line 558 and Figure 12: P4 should refer to 19:00 - 23:00 (not 17:00); please check and correct.

Page 20, Line 567: "12,000" and "37000", I guess they should be "120,000" and "370,000" according to paragraph 5.2 outcomes and figure 11: Correct?

Page 20, Line 569: "Figure 11" I guess should be "Figure 13" here.  

Author Response

Point 1: Your analyses are mainly based on one NTL image dating back to 2011 and social media data of 2010: why did you use data of about 10 years old? Are there available more recent images and data? If yes, do a multi-temporal analysis possibly add a value to your research? Pleas provide some feedback about this.

Response 1: Thanks for the reviewer’s comments. More recent images and social media data are actually available, however, the latest census data in China is just provided in 2010, which is the most important reason why we used the 10 years old data rather than newer data sources. In China, the census survey is conducted every ten years, which also caused most researches focusing on the population mapping in 2000 and 2010. In some large cities, like Beijing, Shanghai and Wuhan, the local government will conduct the survey each year, which makes it possible to do a multi-temporal analysis in these cities. We think it will be very interesting in further studies to adopt 10 years images of these large cities to reveal the pattern of population increasing and consider about how to file the gap between two census surveys in other cities. (Line 172-174, Line 635-637)

Point 2: What is the final resolution of the ISS NTL image after registration with Google Earth? Is it still 25 m? Please clarify.

Response 2: Thanks for the reviewer’s comments. The final resolution of the ISS NTL image after registration was kept unchanged(25m). (Line 182)

Point 3: Equation (2): what NDVI product have you used here? What about spatial resolution of NDVI and co-registration with NTL image? Please provide more detail about.

Response 3: Thanks for the reviewer’s comments. We have added more details in the revised manuscript. The NDVI information was derived from Landsat 7 Enhanced Thematic Mapper Plus (ETM+) images obtained in 2011, and the spatial resolution of NDVI was resampled to 25m (Line 186-190). The spatial resolution of ISS NTL image after registration was set as 25m (Line 241).

Point 4: Figure 3: How the grey area is determined here? Please clarify.

Response 4: The grey area represents the outline of a facility, which was derived from OSM data (Line 267).

Point 5: As a more general comment, I wonder how this approach is portable/reproducible when other cities/urban environments are considered. So, the question is: how much "adaptive" is such an approach to other urban settings? How much is it dependent on data sources (in terms of quality, quantity, spatial/temporal availability, etc.)? For instance, considerations about check-in data volume and acquisition time (see figures 11, 12 and 13) how much do depend on the specific population habits of Wuhan City? Can they be significantly different as far as another city (e.g. in Europe or US) is considered?  Please provide some lines about portability issues in the discussion section.

Response 5: Thanks for the reviewer’s helpful comments.  In the main part of our experiment, we did not distinguish the check-in data of Wuhan according to its volume or acquisition time, which means the population habits of Wuhan City was not considered before the discussion section. So that we think there exists high potential that our approach can be reproduced in other cities. The purpose of discussion data volume is to explain there does not need huge quantity of check-in data in population mapping. The results indicated the approach can be conducted even in small cities without large quantity of Weibo users. The discussion about acquisition time is to investigate how the estimation accuracy can be improved. Although there may exist differences of living habits among cities, most residents will stay at home and relax during the nighttime. Therefore, the conclusion may be suitable for most places, although slight differences may exist in the start or end of the time in other cities. (Line 559-569, 603-609)

Minor issues:

Page 10, Lines 358-359: "Figure 7" I guess should be "Figure 4" here.

Page 14, Line 434: Is "Component 3" (instead of Component 2) here?

Page 19, Line 558 and Figure 12: P4 should refer to 19:00 - 23:00 (not 17:00); please check and correct.

Page 20, Line 567: "12,000" and "37000", I guess they should be "120,000" and "370,000" according to paragraph 5.2 outcomes and figure 11: Correct?

Page 20, Line 569: "Figure 11" I guess should be "Figure 13" here. 

Response 6: Thanks for the reviewer’s pointing out of these issues, all of them have been revised carefully. 

Reviewer 3 Report

Interesting topic, valid approach, and easy to read manuscript. However, I like to draw the attention to few technical and presentation issues.

Concept and Methodology:

I don’t think that Step 4 (Line 213) should be mentioned as a part of the methodology. It is used to validate and discuss the results of the methodology that consists of steps 1 – 3. Similar comment applies with respect to including Step 4 in Figure 2.

The paper is focused on the city of Wuhan and used the related data. I think that the readers can imply the general methodology that can be applied to their cases or cities. However, I wish that the authors had generalized and summarized the methodology more explicitly for the general readership. My concern is, and I wish that the authors had discussed, if this methodology can be applied to other cities but with smaller size and smaller population. In other words, is the success of the methodology depends on having a large data?  

My other concern is that Section 5.2 identifies the appropriate/optimal number of check-in points only after the fact (after applying the methodology). I wish that the authors had come up with a general guideline in this context.

Typos and comments:

Line 92: need to define (use the full name) before using the abbreviations “OLS” and “GWR”.

Line 89, 93 and 100: the use of “(1)”, “(2)”, and “(3)” is redundant considering they were followed by “the first”, “the second”, and “the third” respectively.

Lines 128-138: I’m not sure if the switching to the past tense is appropriate here

Line 154: use “preliminary” instead of “primitively”

Line 171: I don’t think you need to use the word “pairs” when talking about “25 registration points were selected”.

Line 192: use “participants” instead of “cartographers” since we don’t know their qualifications.

Line 306: need to cite the reference to “In the previous study”.

Line 309: need to define (use the full name) before using the abbreviation “RF” for the first time.

Line 324-344: need to cite the reference for the details on the RF method.

Line 358, 359: correct “Figure 7” into “Figure 4”.

Line 440 and 445: add “area” (or “grid”) after “per unit” for clarity. One may assume that “unit” refers to a dwelling unit.

Line 514: add “the community” after “produced by” and before “itself” for clarity.

Author Response

Point 1: I don’t think that Step 4 (Line 213) should be mentioned as a part of the methodology. It is used to validate and discuss the results of the methodology that consists of steps 1 – 3. Similar comment applies with respect to including Step 4 in Figure 2.

Response 1: Thanks for the reviewer's suggestions. The content in the methodology and Figure 2 related to Step4 has been removed.  

Point 2: The paper is focused on the city of Wuhan and used the related data. I think that the readers can imply the general methodology that can be applied to their cases or cities. However, I wish that the authors had generalized and summarized the methodology more explicitly for the general readership. My concern is, and I wish that the authors had discussed, if this methodology can be applied to other cities but with smaller size and smaller population. In other words, is the success of the methodology depends on having a large data? 

Response 2: Thanks for the reviewer’s comments. We have added more words to explain the possibility of repeating our methodology in other cities, especially in small cities.  For Wuhan City, where the population is over 9,000,000(in 2010), the population distribution accuracy can be improved, compared with only NTL data, when the data volume of check-in data is larger than 122,500, which is far less the total check-in data of Wuhan. Our method is conducted based on the similarity matching between POIs data and check-in data, and the most important role for check-in data is to describe the attractive level of POIs. Therefore, the volume of required check-in data is correlated with the number of POIs. For smaller cities with less population and POIs, the estimation accuracy of population mapping can be significantly improved with the adoption of check-in data even in a small quantity. Although the most appropriate data volume of check-in data may be different in other cities due to their different economical structure and urban construction, the results indicated it is appropriate to prepare the check-in data referring to the number of POIs rather than population quantity before analysis (Line 559-569).

Point 3: My other concern is that Section 5.2 identifies the appropriate/optimal number of check-in points only after the fact (after applying the methodology). I wish that the authors had come up with a general guideline in this context.

Response 3: Thanks for the reviewer’s comments. Through the experiment conducted in the discussion section, we can find the most appropriate number of check-in points in Wuhan City is 300,000, which was far less than the total check-in points in Wuhan. So that the meaning for other cities is that there does not need huge quantity of check-in data in population mapping. Moreover, as the method is conducted based on the similarity matching between POIs data and check-in data, the volume of required check-in data is correlated with the number of POIs. So that, the other thing is to keep the magnitude of check-in data according to the number of POIs in other cities (Line 562-569).

Minor issues:

Line 92: need to define (use the full name) before using the abbreviations “OLS” and “GWR”.

Line 89, 93 and 100: the use of “(1)”, “(2)”, and “(3)” is redundant considering they were followed by “the first”, “the second”, and “the third” respectively.

Lines 128-138: I’m not sure if the switching to the past tense is appropriate here

Line 154: use “preliminary” instead of “primitively”

Line 171: I don’t think you need to use the word “pairs” when talking about “25 registration points were selected”.

Line 192: use “participants” instead of “cartographers” since we don’t know their qualifications.

Line 306: need to cite the reference to “In the previous study”.

Line 309: need to define (use the full name) before using the abbreviation “RF” for the first time.

Line 324-344: need to cite the reference for the details on the RF method.

Line 358, 359: correct “Figure 7” into “Figure 4”.

Line 440 and 445: add “area” (or “grid”) after “per unit” for clarity. One may assume that “unit” refers to a dwelling unit.

Line 514: add “the community” after “produced by” and before “itself” for clarity.

Response 4: Thanks for the reviewer’s pointing out of these issues, all of them have been revised carefully.